# EBSD Characterization of the Microstructure of 7A52 Aluminum Alloy Joints Welded by Friction Stir Welding

**DOI:** 10.3390/ma14216362

**Published:** 2021-10-24

**Authors:** Xu Liu, Ruiling Jia, Huixia Zhang, Wenhua Cheng, Xiwei Zhai

**Affiliations:** 1School of Materials Science and Engineering, Inner Mongolia University of Technology, Hohhot 010051, China; 20191100206@imut.edu.cn (X.L.); xiweizhai@163.com (X.Z.); 2State Key Laboratory for Marine Corrosion and Protection, Luoyang Ship Material Research Institute (LSMRI), Qingdao 266101, China; crystalabx@163.com (H.Z.); chengwh@sunrui.net (W.C.); 3School of Mechatronics Engineering, Zhong Shan Polytechnic, Zhongshan 528400, China

**Keywords:** friction stir welding, EBSD, microstructure, Taylor factor, recrystallized grains

## Abstract

The microstructure and texture of materials significantly influence the mechanical properties and fracture behavior; the effect of microstructure in different zones of friction stir-welded joints of 7A52 aluminum alloy on fracture behavior was investigated in this paper. The microstructural characteristics of sections of the welded joints were tested using the electron backscattered diffraction (EBSD) technique. The results indicate that the fracture is located at the advancing side of the thermomechanically affected zone (AS-TMAZ) and the stir zone (SZ) interface. The AS-TMAZ microstructure is vastly different from the microstructure and texture of other areas. The grain orientation is disordered, and the grain shape is seriously deformed under the action of stirring force. The grain size grows unevenly under the input of friction heat, resulting in a large amount of recrystallization, and there is a significant difference in the Taylor factor between adjacent grains and the AS-TMAZ–SZ interface. On the contrary, there are fine and uniform equiaxed grains in the nugget zone, the microstructure is uniform, and the Taylor factor is small at adjacent grains. Therefore, the uneven transition of microstructure and texture in the AS-TMAZ and the SZ provide conditions for crack initiation, which become the weak point of mechanical properties.

## 1. Introduction

7A52 aluminum alloy not only is a 7 xxx aluminum alloy with low density, high ductility, and excellent corrosion resistance, but also has been extensively investigated due to its high strength and hardness [1,2,3]. It has been widely employed as a structural material in aeronautics, military competition, and warships [4,5,6]. When aluminum alloy is welded by traditional welding technology, it is easy to produce welding defects such as porosity, large weld deformation and extensive welding residual stress, resulting in the decrease of mechanical properties [7,8,9,10]. Friction stir welding (FSW) is an emerging solid-state connection method that generates heat through rotation and friction, and plastic pressurizes to form welds, which can effectively reduce or avoid the above defects [11,12]. Friction stir-welded joints have different heat input and stirring force in the welding process; the welded joints are divided into the heat-affected zone on the advancing side (AS-HAZ), the thermomechanically affected zone on the advancing side (AS-TMAZ), the stir zone (SZ), the heat-affected zone on the retreating side (RS-HAZ), and the thermomechanically affected zone on the retreating side (RS-TMAZ). Lan Zhang et al. [13] have reported on the microstructural and mechanical evolution of the FSW 6061-T6 aluminum alloy, where the grain size, hardness, and impact-absorbing energy of the stir zone was smaller than those of the heat-affected zone. W.F. Xu et al. [14] found that in microstructure evolution during the conventional single-side friction stir welding (SS-FSW), significant grain refinement and a larger percentage of high-angle grain boundaries (HAGBs) exist in the SZ and TMAZ; the fracture location is TMAZ. In summary, the microstructure of friction stir-welded joints in different zones is different, and the cracking behavior and mechanical properties of welded joints are distinct [15,16,17,18].

An elaborate and complete description of the microstructure of a crystalline material should include knowledge about crystallographic orientation features and textures of the constituent grains, and the distribution of constituent phases [19,20]. Many ways of characterizing the microstructure of crystals usually reveal that the material microstructure can be given by grain orientation image mapping (OIM), orientation distribution function (ODF), grain boundary angle, grain size, and Taylor factor, with a variety of graphs and quantitative relationship graphs [21,22,23]. However, the effect of microstructure and texture in different zones of 7A52 aluminum alloy FSW joints on mechanical properties are still unclear. Therefore, this paper mainly presents the following: (1) microstructure and texture analysis of friction stir-welded joint of the 7A52 aluminum alloy; (2) the effect of microstructure on mechanical properties and causes of cracking. Furthermore, the correlation between microstructure and mechanical properties is revealed.

## 2. Materials and Experimental

The 7A52 aluminum alloy is a rolled plate provided by the Northern Heavy Industry Group (Shenyang, China). The size of the base material is 150 mm × 400 mm × 12 mm; the main chemical composition is shown in Table 1. The tensile strength of the plate is 493 MPa, the yield strength is 427 MPa and the elongation is 11%. The friction stir-welding parameters are the rotation rate of 500 rpm; traverse speed of 80 mm/min; the angle between the stirring head and the main shaft of the welding machine is 2.5°; and the pressing amount of the stirring tool is 0.2 mm, butt-welded by FSW perpendicular to the rolling direction (RD). After welding, a two-stage solution heat treatment, 460 °C/120 min + 480 °C/60 min was adopted. The test of mechanical properties used a WDW-200 electronic testing machine, and the instrument is produced by Changchun Kexin Testing Instrument Co., Ltd. in Jilin Province, China; three identical specimens were used for parallel tests in each group, the tensile rate was 1 mm/min [24,25].

The sample for electron backscattered diffraction (EBSD) observation was taken perpendicular to the welding direction (WD), which used Wire-Cut Electrical Discharge Machining, Produced in Taizhou Ruite Machinery Equipment Co., Ltd., Jiangsu, China. The test zone of EBSD was selected as the middle part of the welded joint, then electropolished technology processes samples were used, with a solution of 30 mL HNO_3_ (made in Hohhot, China) + 60 ml CH_3_OH at an operation voltage of 20 V for 70 s [26,27], keeping the temperature at −20 °C~−30 °C. Fracture morphology and the specimen of EBSD were characterized using a Scanning Electron Microscope ( Quanta 650 FEG, made in FEI, Hillsboro, OR, USA )) equipped with an EBSD detector. The step size of the EBSD scanning was 0.6 μm, and the acceleration voltage of the SEM is 20 kV for the different zones of welding (AS-TMAZ, SZ, RS-TMAZ). Data analysis was performed on the HKL Channel 5 software (made in HKL Technology, Hobro, Denmark).

## 3. Results and Discussion

### 3.1. Cross Section of Welded Joint

The cross section of the 7A52 aluminum alloy FSW joint is shown in Figure 1a. We can see that the SZ is “V”—shaped, the shaft shoulder is broader and the bottom is narrower, and two dividing lines are on both sides of the SZ. The obvious boundary is AS-TMAZ, and the blurred boundary is RS-TMAZ; this is caused by the different flow rates of plastic metal on both sides of the weld nugget during the welding process. On the advance side where the welding direction is the same as the rotation direction of the stirring tool, and where the speed gradient of the plastic metal is more significant, the retreating side is the opposite. Between TMAZ and the base material is HAZ, the aggregation state of the second phase and grain structure at the different regions have changed greatly. The second phase of AS (Figure 2b) is mainly distributed in strip shape, where it is easy to produce grain cracking. A small stirring force and large velocity gradient difference lead to the lack of smooth transformation and uneven structure, and its mechanical properties are weakened. The second phase particles are dispersed in the SZ (Figure 2c), and the microstructure is subject to strong stirring and heat input, and the equiaxed crystals are formed. The second phase particles of the RS (Figure 2d) are distributed sporadically, the microstructure is less affected by the particles, and the deformation of grains mainly comes from heat input.

### 3.2. Fracture

Figure 2a shows the fracture positions of three groups of parallel specimens. It can be seen that the fractures are located at the SZ–AS interface, and it is concluded that the mechanical properties here are the weakest. The mechanical property curve of the welded joint is shown in Figure 3b. It can be seen that the toughness of the test piece is small and there is no obvious neck rope phenomenon; the tensile strength is 369 MPa, the yield strength is 279 MPa and the elongation is 7%. The fracture modes of the shaft shoulder, middle and bottom of the friction stir-welded joint are different. Figure 2c shows the bottom of the fracture. There are many dimples, which are pits left by the direct falling off of the second phase and the matrix, which cannot better hinder the crack propagation and crack directly along the interface and grain boundary. The middle of the fracture (Figure 2d) has large deformation. The dimples were a feature of ductile fracture, the second phase in the dimple was tested by EDS and found to be Mg_2_Si and AlFe, respectively, which occurred from the second phase inside, and acted as a barrier for crack propagation, when the particles cannot resist external deformation, the second phase breaks, resulting in the deformation of the surrounding grain structure, and those grain structures with large deformation and uneven transition have become the preferred path of crack propagation, as previously reported by M.R. Jandaghi [28] and R. Beygi [29] et al.

### 3.3. Grain Orientation

The grain orientation of different regions along the weld direction was characterized by EBSD technology, as shown in Figure 3. From the orientation maps, an inhomogeneous microstructure and no preferred crystallographic orientation in the FSW welding joint structure are seen, due to the rearrangement of grain structure after welding, resulting in different microstructures. The SZ (Figure 3b) and RS-TMAZ (Figure 3c) are dominated by < 111 > crystal orientation, but the AS-TMAZ (Figure 3a) grain orientation is not obvious. Figure 3e,f show the orientation of grains at the same position in the yield and fracture of the RS. The grain orientation of both is <111>, but we can see that when the welded joint is subjected to external force, the grains will produce the preferred orientation. When the initial grain orientation is inconsistent with the final grain orientation, the grains will rotate, and when the crystals are in the process of rotation, they will be hindered by the second phase particles and grain boundaries, resulting the accumulation of dislocation events, which easily becomes the location of crack initiation. If the initial grain orientation is consistent with the final grain orientation, the grain will be damaged. The rotation of the AS-TMAZ grains with nonuniform grain orientation is the most obvious and reduces the mechanical properties.

### 3.4. Grain Shape and Grain Size

The grain size distribution of different weld zones with FSW welding process is shown in Figure 4, the change range of color is shown in the subscript in the figure, gradually increasing from blue to red. The grain size of the FSW joint at the AS-TMAZ (Figure 4a) was more inhomogeneous and more extensive. The SZ (Figure 4b) presents the grain size as smaller and with an “onion ring” structure; the grain size of RS-TMAZ (Figure 4c) was homogeneous, the gap between the grain size of the RS-TMAZ and SZ is small, but the grain size is elongated by stirring force.

From the AS-TMAZ to SZ to RS-TMAZ, the average grain size decreases first and increases. The average grain size appears to be 13.51 μm in the SZ, and AS-TMAZ has the largest grain size of 15.46 μm. The average grain size of the TMAZ is larger than the SZ. The small average grain size will bring more grain boundaries, better restraining the effect on the movement of dislocations, and resulting in a low effective heat input. Only sufficient flow of the SZ can be ensured during the welding process, while the flow rate of the TMAZ drops very rapidly, which eventually leads to significant changes in the structure between the weld nugget zone and the thermomechanically affected zone. It can be seen from the grain size of the AS-TMAZ that the grain size has an aggregation phenomenon; the larger grains gather (yellow box) and the smaller grains assemble (black box). The grain size is not gradual but abrupt. Therefore, the AS-HAZ, with large average grain size, is easy to crack and becomes a weak mechanical property. 

### 3.5. Grain Boundary

The adjacent grain boundary with a phase difference between 2° and 15° is called a low-angle grain boundary (LAGBs), and the adjacent grain with a phase difference of more than 15° is called a high-angle grain boundary (HAGBs). The grain boundaries maps of the 7A52 aluminum alloy FSW joint were displayed in Figure 5, HAGBs were represented by the green line, and LAGBs were represented by the red line. The grain boundary in AS-TMAZ (Figure 5a) is more confusing than the SZ (Figure 5b) and the RS-TMAZ (Figure 5c). Owing to the transition region between the SZ and the TMAZ, there is significant plastic deformation caused by the rotating of the mixing tool and shear deformation around the material of the SZ; grain boundaries are irregularly distributed at welded joints, with some differences in LAGBs and HAGBs. The percentage of LAGBs is 36% and 44% for the AS-TMAZ and RS-TMAZ, respectively, and the proportions are lower than the weld nugget zone, which is 56%. The atoms at the grain boundary are arranged irregularly; thus the existence of the grain boundary will hinder the movement of dislocation at room temperature, resulting in the improvement of plastic deformation resistance. The macro performance is that the grain boundary has high strength. The average grain boundary angle percentage is 26.85°, 18.99°, and 24.32° for the AS-TMAZ SZ and RS-TMAZ, and the average grain boundary angle in the weld nugget zone is smaller than AS-TMAZ and RS-TMAZ. There is a significant difference between the AS-TMAZ and SZ, in that when the material is subjected to external force, the grain boundary will change to resist the deformation of external force. The uneven transition will cause an abrupt change in the ability to resist deformation, leading to a decrease in mechanical properties. Thus, the mechanical properties of the AS-TMAZ are relatively weak.

### 3.6. Taylor Factor

The Taylor factor refers to the ability of polycrystalline materials to resist plastic deformation. The more significant the Taylor factor, the stronger the ability to resist plastic deformation, and the smaller the Taylor factor, the weaker the ability to resist plastic deformation. At the same time, it is also closely related to the external stress field function distribution. In addition, if there is a large difference in the Taylor factor between adjacent microstructures or adjacent grains. Where the ability of the crystals to resist plastic deformation is not coordinated, stress concentration will occur, which will easily lead to the initiation of cracks. The Taylor factor maps of the 7A52 aluminum alloy FSW joint are displayed in Figure 6, and the change range of color is shown in the subscript in the figure, gradually increasing from blue to red. It can be found that the AS-TMAZ (Figure 6a) is most significantly influenced by the Taylor factor; the influence of the Taylor factor of the SZ (Figure 6b) is the smallest. At the AS-TMAZ–SZ interface and inside the AS-TMAZ grains, the Taylor factor has an enormous difference. However, the difference of the Taylor factor in the RS-TMAZ (Figure 6c) and RS-TMAZ–SZ interface is small. Therefore, the AS-TMAZ–SZ interface is not uniform, and the transition provides conditions for crack initiation, which reduces the mechanical properties and causes a cracked zone.

### 3.7. Effect of the Welding Process on Recrystallization Behavior

Figure 7 shows the morphology of recrystallized grains, substructure grains and deformed grains in different zones of the 7A52 aluminum alloy FSW joint, and corresponding percentages of different crystal grains are shown in Figure 7d. The blue region represents recrystallized grains, the red region represents deformed grains, and the yellow part represents substructured grains. It is found that the deformed grains are mainly tiny grains sporadically distributed in each region. The distribution of the substructured grain has an apparent aggregation state. From the AS-TMAZ to SZ to RA-TMAZ, the percentage curve of recrystallized grains presents a similar “V” shape. The percentage of recrystallized grains occupied by the SZ (Figure 7b) is the smallest, which is 40.96%, and the AS-TMAZ (Figure 7a) is 75.56% and RS-TMAZ (Figure 7c) is 74.15%; the percentage curve of substructured grains and deformed grains present an inverted “V” shape. The maximum percentage of the substructured grains and deformed grains reaches 45.76% and 13.28% for the SZ and AS-TMAZ. Substructured grains and deformed grains are slightly lower than the RS-TMAZ, at 19.89% and 4.54%, respectively.

In the process of friction stir welding, the welded joint undergoes violent stirring and high-temperature thermal cycling by the stirring tool, and dynamic recrystallization occurs in the weld nugget zone, and the plasticity increases. Because the welding material is a thick plate, the heat dissipation is slow in the cooling process so that the originally grown dynamic recrystallized grains further grow and produce a large number of substructured grains [30,31].

In the welding process, the structure of the heat-engine affected zone is affected by both the mechanical stirring of the stirring needle and the thermal welding cycle. However, because the TMAZ is far away from the stirring tool, the TMAZ is not as affected by plastic deformation and temperature changes within the SZ, resulting in a greater degree of bending deformation of the grain structure in the region, and a recovery reaction occurs under the action of thermal cycling; a recovery grain structure is formed in the lath structure [32].

## 4. Conclusions

This paper mainly discusses the influence of friction stir welding on the welding process of 7A52 aluminum alloy; it is found that the cracking behavior is caused by different microstructures. Based on the results obtained in the study, the following conclusions are drawn: The 7A52 aluminum alloy FSW fracture is located at the junction of AS-TMAZ–SZ, with many dimples in the it, and it is a ductile fracture.The SZ and RS-TMAZ are dominated by < 101 > crystal orientation, the grain size is small, and a low-angle grain boundary accounts for the majority of boundaries, effectively hindering the movement of dislocations. The Taylor factor distribution in the weld nugget zone is relatively uniform. The microstructures of the SZ and RS-TMAZ are uniformly transitioned without stress concentration.The AS-TMAZ has serious grain structure deformations. The grain orientation is disorderly, a high-angle grain boundary accounts for a relatively high proportion of boundaries, the Taylor factor has a large difference, and the resistance to external force deformation is poor. Therefore, the abrupt grain structure in the region leads to the uneven transition of the microstructure, and this becomes the weak point of its mechanical properties.The TMAZ microstructure is dominated by discontinuous dynamic recrystallization; the SZ accounts for less, but the substructured grain shows an inverse relationship. The degree of recrystallization is mainly controlled by the stirring force in different directions.

## Figures and Tables

**Figure 1 materials-14-06362-f001:**
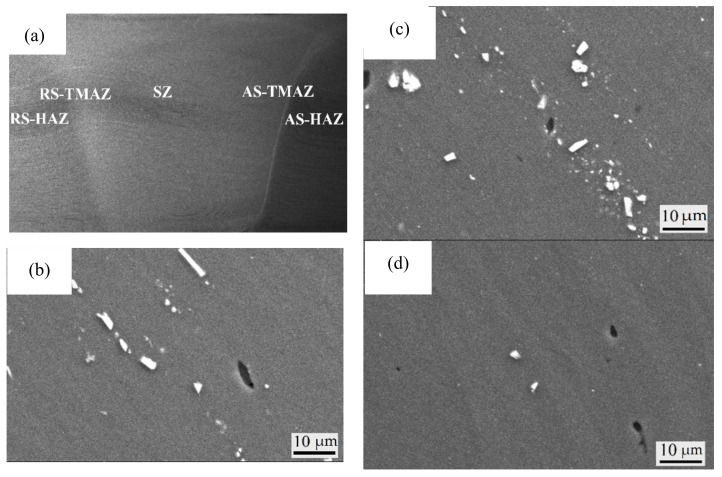
The second phase particles of 7A52 aluminum alloy FSW joint: (**a**) The cross section of the 7A52 aluminum alloy FSW joint; (**b**) AS-TMAZ; (**c**) SZ; (**d**) S-TMAZ 7A52 aluminum alloy FSW joint fracture organization.

**Figure 2 materials-14-06362-f002:**
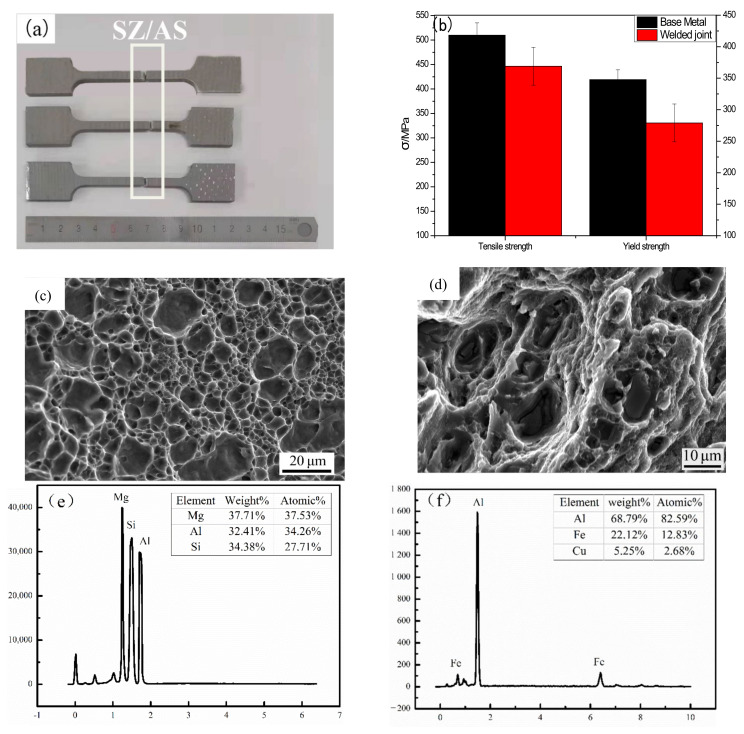
Mechanical properties of FSW joint of 7A52 aluminum alloy (**a**) Fracture location; (**b**) stress–strain curve; (**c**,**d**) 7A52 aluminum alloy FSW joint fracture organization; (**e**) EDS analysis of Mg2Si; (**f**) EDS analysis of AlFe.

**Figure 3 materials-14-06362-f003:**
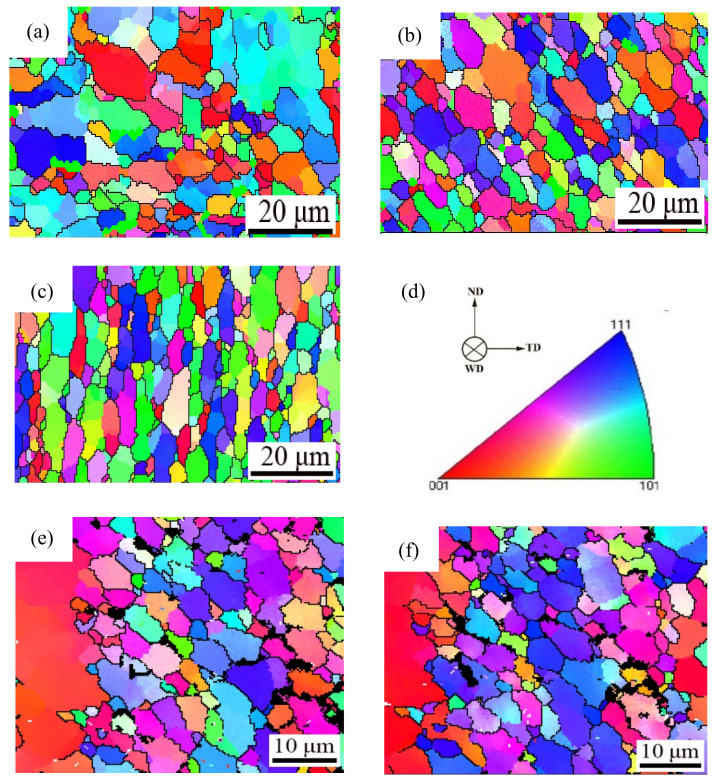
Grain orientation maps: (**a**) AS-HAZ; (**b**) SZ; (**c**) RS-TMAZ; (**d**) Orientation standard of 7A52 aluminum alloy FSW joint; (**e**) Yield of the RS; (**f**) Fracture of the RS.

**Figure 4 materials-14-06362-f004:**
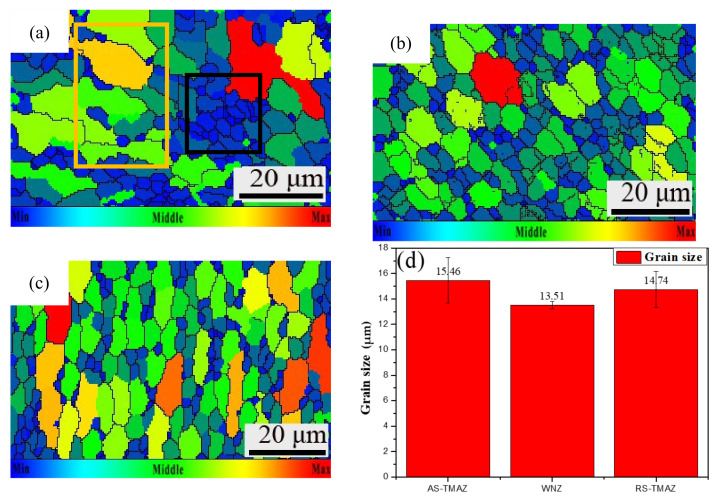
Grain size maps: (**a**) AS-HAZ; (**b**) SZ; (**c**) RS-TMAZ; (**d**) Value of grain size of 7A52 aluminum alloy FSW joint.

**Figure 5 materials-14-06362-f005:**
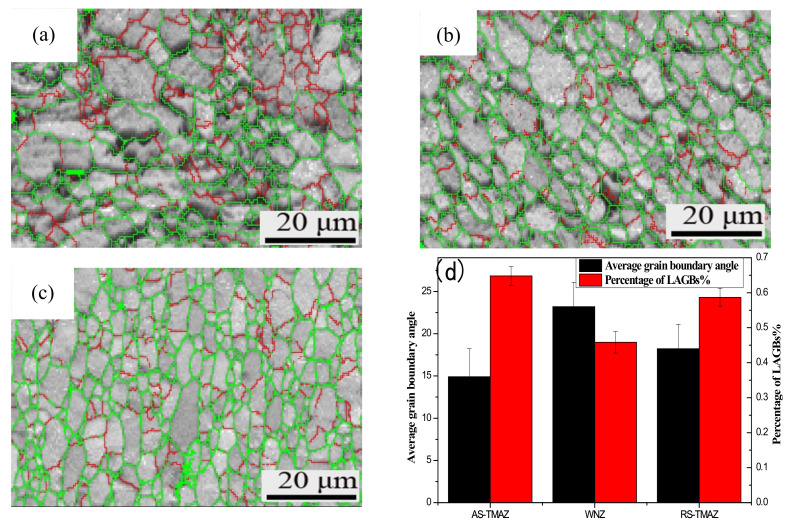
Grain boundary maps: (**a**) AS-HAZ; (**b**) SZ; (**c**) RS-TMAZ; (**d**) Value of grain boundaries of 7A52 aluminum alloy FSW joint.

**Figure 6 materials-14-06362-f006:**
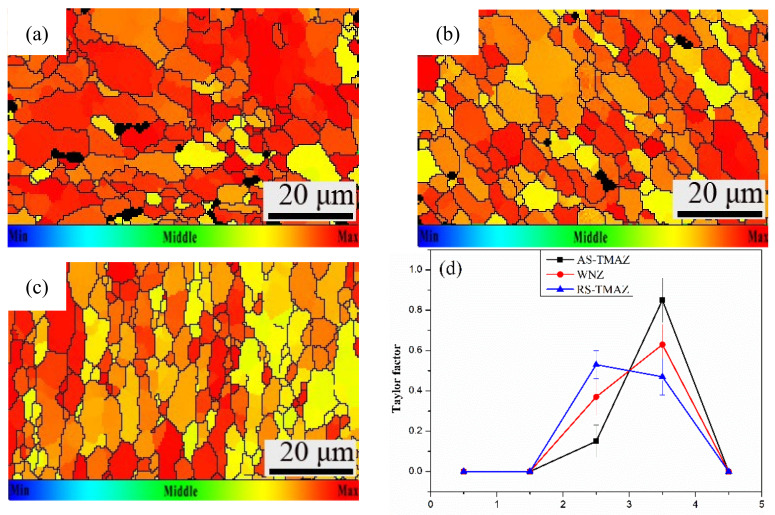
Taylor factor maps: (**a**) AS-HAZ; (**b**) SZ; (**c**) RS-TMAZ; (**d**) Value of Taylor factor of 7A52 aluminum alloy FSW joint.

**Figure 7 materials-14-06362-f007:**
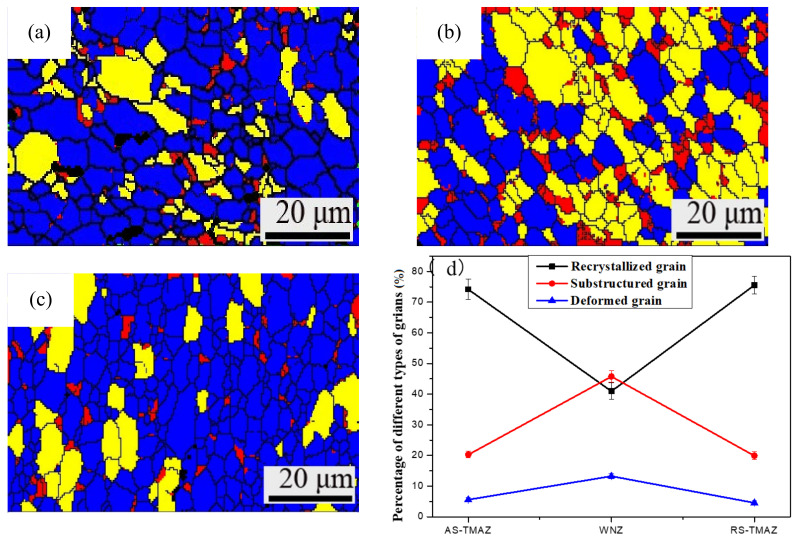
Grain recrystallization behavior maps: (**a**) AS-HAZ; (**b**) SZ; (**c**) RS-TMAZ; (**d**) Value of grain morphology of 7A52 aluminum alloy FSW joint.

**Table 1 materials-14-06362-t001:** 7A52 aluminum alloy main chemical composition (mass fraction/%).

Base Metals	Mg	Zn	Mn	Cr	Ti	Cu	Fe	Si	Al
7A52	2.40	4.20	0.35	0.20	0.12	0.12	<0.30	<0.25	Bal

## Data Availability

Data sharing is not applicable to this article.

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
