# Peer review of "EBSD Characterization of the Microstructure of 7A52 Aluminum Alloy Joints Welded by Friction Stir Welding"

_materials, 2021, doi:10.3390/ma14216362_

Round 1

Reviewer 1 Report

However, the title of this paper is interesting, but there are many problems in the manuscript that must be corrected to reach acceptable quality for publication.

1- The language of this paper is catastrophic. I advise the authors to ask a native person to review their work and correct the grammatical errors. There are many sentences hard to understanding in the passage.
2- Please change the code of Welding nugget zone (WNZ) to a known abbreviation like Stir zone (SZ) or Nugget. In the current style, it is confusing for the readers who know the FSW process very well.
3- The authors showed the fracture surface in Fig. 1. But I could not find the results of the mechanical test. Please add the stress-strain curve of the sample. Likewise, the analysis of the fracture surface was very poor. The failure point under the tensile loading is not indicated. Please expand your explanation by finding the relation between the failure point and the microstructural features and comparing the obtained results with the reported observations in the literature. To get an idea about how to analyze the fracture surface of your samples, please review the below papers and add them to your reference list: “Journal of Manufacturing Processes, Volume 70, October 2021, Pages 152-162”, “Journal of Manufacturing Processes, Volume 57, September 2020, Pages 712-724”.  
4- The presented discussion in Fig. 2 does not make sense. Please discuss with somebody who is an expert in EBSD analysis and revise this section. when you talk about the grain’s orientation, you must bring the reasons for the preferential orientation toward a certain direction. In the meantime, again the authors attributed the drop in mechanical strength to the grain’s orientation. But which mechanical test in which direction? Along the weld line or perpendicular to it?
5- Fig. 4 is disordered. It is not possible to check the presented graphs. Please put the images in a template and transfer the collection to the word file to avoid this problem. Fig. 5 has also a similar problem.
6- EBSD micrographs of Figs. 4, 5, and 6 need to have a label. In the current format, the variation range of the colors is not mentioned.
7- There is an essential lack of SEM analysis from different phases that can form in different welding zones. It is possible that the failure point is enriched by some intermetallic particles. So please provide the SEM images and related EDS and MAP analysis from the particles formed in different zones, particularly near the failure point.

Reviewer 2 Report

This manuscript describes a study on the microstructural zones in Friction stir welding of aluminium alloy 7A52. 

First of all, the overall level of English needs to improve. For example, it should read 'Taylor factor' and not 'tayor factor', throughout.

There are major shortcomings with the work that need to be addressed before I can recommended publication.

1) Was the material heat treated in any way and what are the expected mechanical properties of the base material.

2) We see only one image of a fracture surface, which is insufficient to describe the fracture of all of the test bars. Average strength (with error bars) should be presented and compared to the base material. Images should also show the location on the fracture with respect to the part and the weld zone. Repeat tests should have been done to generate statistics and to show the recurrence of the fracture at the same location.

3) All of the data is presented without error bars. For example, in figure 3(d), we cannot tell if the differences in grain size are significant. I would recommend error bars (standard error) and a significance test (t-test or similar) between the groups to see if the differences in grain size are significant. This comment applies to all of the data presented on each figure.

4) Figure 4 is badly laid out on the page. I cannot see the graph in my version.

Round 2

Reviewer 1 Report

The revised manuscript could be considered for publication.

Reviewer 2 Report

Overall, I feel that the required changes have been made. Fig 6(d) stills says Tayor instead of Taylor. Therefore, I think that the manuscript requires minor changes to the level of English